# Towards Decomposed Linguistic Representation with Holographic Reduced Representation

## Abstract

The vast majority of neural models in Natural Language Processing adopt a form of structureless distributed representations. While these models are powerful at making predictions, the representational form is rather crude and does not provide insights into linguistic structures. In this paper we introduce novel language models with representations informed by the framework of Holographic Reduced Representation (HRR). This allows us to inject structures directly into our word-level and chunk-level representations. Our analyses show that by using HRR as a structured compositional representation, our models are able to discover crude linguistic roles, which roughly resembles a classic division between syntax and semantics.[1]

## 1 Introduction

Recent advances in representation learning have been unequivocally led by the long strides of progress in deep learning and its distributed representations. In many tasks of Natural Language Processing (NLP), researchers have convincingly shown that distributed representations are capable of encoding the complex structure of textual inputs (for example Mikolov et al. (2010; 2013); Józefowicz et al. (2016); Sutskever et al. (2014)). The dominant approach for many NLP tasks is the encoder-decoder paradigm that uses neural networks to learn the transformations from many smaller comprising units to one complex embedding, and vice versa [2]. The underlying structure, in a rather crude fashion, is assumed to be represented by this complex embedding. In many cases, such crude way of representing the structure is unsatisfactory, due to a lack of transparency, interpretability and transferability. Transparency and interpretability require that the operations of encoding and decoding have clear conceptual meaning, and transferability generally necessitates a separation of transferable features (e.g., domain-invariants) from the rest. On account of shortcomings, much previous work has been devoted to inducing disentangled representations (Chen et al., 2016; Hsu et al., 2017; Higgins et al., 2016; Janner et al., 2017).

We attempt to address these issues by utilizing a more principled framework to encode complex symbolic structures using distributed representations. Specifically, we employ Holographic Reduced Representation (HRR) to represent and manipulate structures. As a member of the Vector Symbolic Architecture (VSA) family (Gayler, 2003; Smolensky, 1990; Plate, 1995; Kanerva, 2009), HRR builds upon the notions of roles and fillers (i.e., values for the roles). For instance, with semantic roles, the sentence *John loves his mom* can be represented by three role-filler pairs, namely (`agent`, *John*), (`predicate`, *loves*), and (`patient`, *his mom*). Each role and filler is represented by a high-dimensional vector, and HRR provides a mathematical framework to encode role-filler pairs, compose complex embeddings, and retrieve fillers given corresponding roles. A disentangled representation, using HRR terminology, is synonymous with decomposing a complex embedding into many role-filler pairs.

In this paper, we investigate the effectiveness of HRR at inducing disentangled representations on the task of language modeling (LM). We applied HRR to language modeling because it requires

---

[1] Code will be released to the public.

[2] For attention mechanisms, there is actually a distribution over multiple embeddings, rather than one single complex embedding.

minimal supervision, and has been proven hugely beneficial for many other NLP tasks. The versatility of language modeling demonstrates that some linguistic regularities much be present, and the training signal is sufficient for them to arise. We carefully design a language model with HRR that explicitly encodes the underlying structure as role-filler pairs on both word-level and chunk-level, and show that HRR provides an inductive bias towards the learning of decomposed representations. We demonstrate that on both Penn Treebank (PTB) and a subset of One-Billion-Word LM data set (1B), our model can effectively separate certain aspects of word or chunk representation, which roughly corresponds to a division between syntax and semantics. We perform various analyses on the learned embeddings, and validate that they indeed capture distinct linguistic regularities.

Our paper is structured as follows. Section 2 gives a background overview of VSA and HRR; Section 3 details our proposed models; Experimental results are shown in Section 4, followed by related work in Section 5 and a conclusion in Section 6.

## 2 BACKGROUND

Vector Symbolic Architecture (VSA) is a family of models that enable connectionist models to perform symbolic processing, while encoding complex structures in distributed representations. A set of algebraic operations defined by these approaches allow them to compose, decompose and manipulate symbolic structures.

Our paper focuses on such an approach, namely Holographic Reduced Representation (HRR) proposed by (Plate, 1995). HRR uses three operations: circular convolution, circular correlation and element-wise addition, to perform encoding, decoding and composition, respectively. The cicircular convolution (denoted by the operator $\circledast$) of two vectors $\mathbf{x}$ and $\mathbf{y}$ of dimension $d$, is defined as $\mathbf{z} = \mathbf{x} \circledast \mathbf{y}$, in which

$$z_i = \sum_{k=1}^{d} x_{i \bmod d} \, y_{(i-k) \bmod d}, \quad i = 1 \ldots d$$

$\mathbf{x} \circledast \mathbf{y}$ is called the binding of $\mathbf{x}$ and $\mathbf{y}$, or the encoding of the pair $(\mathbf{x}, \mathbf{y})$. Using this operation, the composition of a set of role-filler pairs $(\mathbf{r}_1, \mathbf{f}_1), (\mathbf{r}_2, \mathbf{f}_2), \ldots, (\mathbf{r}_m, \mathbf{f}_m)$ is represented as $\mathbf{H} = \mathbf{r}_1 \circledast \mathbf{f}_1 + \mathbf{r}_2 \circledast \mathbf{f}_2 + \ldots \mathbf{r}_m \circledast \mathbf{f}_m$, where + is element-wise addition used as a composition operator. The previous example *John loves his mom* can be represented as

$$\mathbf{r}_{agent} \circledast \mathbf{f}_{John} + \mathbf{r}_{predicate} \circledast \mathbf{f}_{loves} + \mathbf{r}_{patient} \circledast \mathbf{f}_{his \ mom}.$$

By definition, HRR guarantees that the composed representation remains a vector of dimension $d$, regardless of how many items are bound together by the $\circledast$ operation. This avoids the parameter explosion problem of other VSA approaches such as as Tensor Product Represenation (TPR) (Smolensky, 1990), and makes HRR a more practical choice for representing compositional structures.

To decode from an HRR and retrieve a role or filler representation, an approximate inverse operation of the circular convolution, named circular correlation, is defined as $\mathbf{t} = \mathbf{x} \oplus \mathbf{y}$ in which

$$t_i = \sum_{k=1}^{d} x_{i \bmod d} \, y_{(i+k) \bmod d}, \quad i = 1 \ldots d \tag{1}$$

Now given a memory trace $\mathbf{z} = \mathbf{x} \circledast \mathbf{y}$, the correlation operation allows us to retrieve $\mathbf{y}$ from the cue $\mathbf{x}$ via $\mathbf{y} \approx \mathbf{x} \oplus \mathbf{z}$. We do not detail the exact conditions when this retrieval holds, but refer interested readers to the original paper (Plate (1995))

## 3 HOLOGRAPHIC REDUCED REPRESENTATION FOR LANGUAGE MODELING

We incorporate HRR into language models on two levels: word level and chunk level. Our HRR-enabled language models (HRRLM) posit an explicit decomposition of word or chunk representations, which enables our model to capture different aspects of linguistic regularities. Before delving into the details of our model, we first introduce notations and provide a brief account of the commonly used RNN-based LM.

### 3.1 RNNLM

RNN-based LMs estimate the probability of any given sentence $s = w_1 w_2 \ldots w_n$ using an RNN. At each step $t$, RNN encodes the history $w_1 w_2 \ldots w_t$ into a vector $h$, and tries to predict the next token $w_{t+1}$. Prediction is generally modeled by a linear layer followed by softmax operation. Specifically,

$$\Pr(w_t | w_1, \ldots, w_{t-1}) = \frac{\exp(score(h, E(w_t)))}{\sum_{w' \in V} \exp(score(h, E(w')))}, \tag{2}$$

$$score(h, E(w)) = h \cdot E(w), \tag{3}$$

where $E(\cdot)$ is the embedding operation to embed a symbol into a continuous space $\mathbb{R}^d$, and $V$ is the entire or a sampled subset of the vocabulary. The scoring function is defined by dot product, and therefore maximizing this probability encourages related words to form clusters, and away from other words in the embedded space.

### 3.2 WORD-LEVEL HRRLM

**Encoding**  We first use HRR to directly encode the underlying structures of words. We assume there is a decomposition of representations along $N$ directions. Specifically, we embed a word $w$ as

$$\tilde{E}(w) = \sum_{i=1}^{N} \mathbf{r}_i^{word} \circledast \tilde{E}_i(w), \tag{4}$$

where $\mathbf{r}_i^{word}$'s are *basis* role embeddings, shared by all words. Each basis role embedding is bound to its distinct set of filler embeddings, modeled by $\tilde{E}_i$. The motivation is that when properly trained, different bindings should capture disparate aspects of word representation. For instance, the first binding might be relevant for syntactic categories, and the second one for semantic relatedness. With this particular decomposition, the word *getting* should be close to other gerunds such as *giving* and *forgetting* in the first embedding space, and *get*, *got* or *received* in the second. In this case, the composite (i.e., the sum) of these bindings essentially encodes $getting = \{\texttt{semantics}: \text{GET}, \texttt{syntax}: \text{GERUND}\}$.

We additionally assume that each set of filler embeddings $\tilde{E}_i$ resides in a separate linear subspace. This is achieved by modeling each $\tilde{E}_i(w)$ with a linear combination of its associated *basis* filler embeddings $\mathbf{f}_{i,j}$. Specifically,

$$\tilde{E}_i(w) = \sum_{j=1}^{d'} s_{i,j}^w \mathbf{f}_{i,j} = [\mathbf{f}_{i,1}; \mathbf{f}_{i,2}; \ldots; \mathbf{f}_{i,d'}] \begin{bmatrix} s_{i,1}^w \\ s_{i,2}^w \\ \vdots \\ s_{i,d'}^w \end{bmatrix} = \mathbf{F}_i s_i^w$$

where $s_i^w \in \mathbb{R}^{d'}$ is a word-specific $d'$-dimensional vector. In other words, $\mathbf{F}_i$ projects $s_i^w \in \mathbb{R}^{d'}$ to $\tilde{E}_i(w) \in \mathbb{R}^d$. This assumption has two advantages. First, the total number of parameters for the embedding layer is now $VNd'$. We can set a smaller value $d'$ to prevent overparameterization, while maintaining a $d$-dimensional vector as the input to RNN. Second, by having separate bases for different role-filler bindings, we introduce an inductive bias for the model to learn a decomposition of word representation. This requirement makes sense intuitively – a decomposition of representation usually necessitates a separation of feature space. Our preliminary experiments show that this separation is essential for obtaining decomposed representations. The entire encoding operation is illustrated in the bottom half of Figure 1(b).

**Decoding**  The composite embedding $E(w)$ is fed as input to RNN. From its output $h$ given by the top hidden layer, we decode all the filler vectors using circular correlation, and factorize the scoring function into $N$ parts. Specifically, we use the same word level loss as in Equation (2), but replace the scoring function in Equation (3) as a sum of dot products in every filler space:

$$f_i = \mathbf{r}_i^{word} \oplus h,$$

$$score(h, w) = \sum_{i=1}^{N} \alpha_i \Big[ f_i \cdot \tilde{E}_i(w) \Big] \tag{5}$$

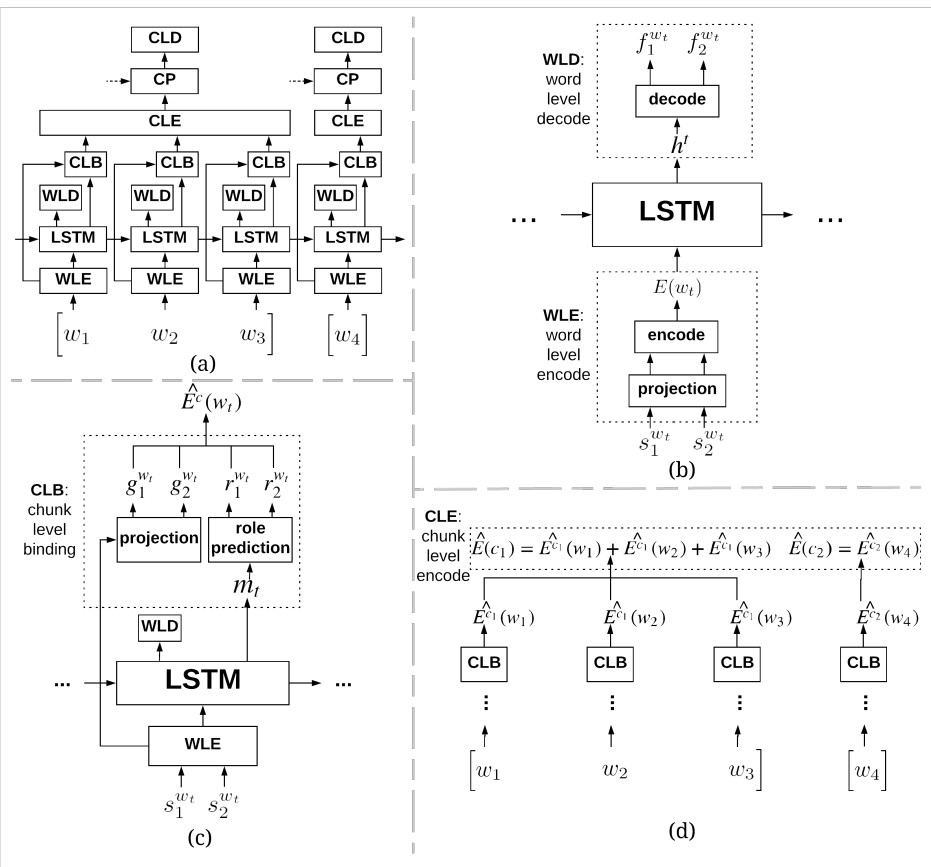

Figure 1: Architecture of HRRLMs. (a) Chunk-level HRRLM. (b)Word-level HRRLM. (c) Step 1 and 2 of chunk-level model encoding. (d) Step 3 of chunk-level model encoding. See Section 3 for details.

where $\alpha_i$'s are scalar hyperparameters that we use to break the symmetry of the scoring function. Specifically when $N = 2$, $\alpha_1$ is set to a constant 1.0, and $\alpha_2$ is linearly annealed from 0.0 at the start of training to 1.0 at a specified time step $T$, then remains constant afterwards. Note that dot products are only computed between co-indexed filler embeddings, namely between $f_i$ and $\tilde{E}_i(w)$. This ensures the model only learns relatedness in the $i$-th subspace, without interference from other subspaces. The entire word-level HRRLM is illustrated is Figure 1(b).

**Regularization on basis embeddings**    Basis embeddings are chosen so that they are not correlated with each other. This decorrelation is promoted by adding an isometric regularization term to the basis embeddings. For instance, we define an isometric regularization penalty for the basis fillers $\mathbf{F}_i$'s as:

$$\Phi(\mathbf{F}_1, \mathbf{F}_2, \ldots, \mathbf{F}_N) = \beta(\|\mathbf{F}_i^\top \mathbf{F}_i - \mathbf{I}\|^2 + \|\mathbf{F}_i^\top \mathbf{F}_j\|^2), \quad \forall 1 \leq i, j \leq N, i \neq j,$$

where $\mathbf{I}$ is an identity matrix, and $\beta$ is a hyperparameter. Similarly, we add a regularization term for $\mathbf{r}_i^{word}$'s. As an alternative, we also consider using fixed random vectors for basis embeddings since high-dimensional random vectors are approximately orthogonal to each other.

### 3.3    CHUNK-LEVEL HRRLM

A direct extension to the chunk-level model using the proposed techniques above is not desirable, due to two major difficulties. First, Equation (4) stipulates that each unique word type is assigned a vectorial parameter, which is computationally infeasible due to the vast number of unique chunks. Second, the same chunk can carry different semantic roles. For instance, the two sentences *His mom*

*loves John* and *John loves his mom* have the same noun phrases, but with their roles of agent and patient switched. In light of these difficulties, we take on a compositional approach to address the first issue, and use a context-sensitive role to address the second.

**Encoding**   Unlike the word-level HRR representation in Equation (4) where roles are fixed, the chunk roles are represented by linear combinations of $M$ *basis* role embeddings. This formulation allows us to model context-dependency. Specifically, we construct HRR representations for a chunk $c = w_1 w_2 \ldots w_m$ in three steps:

STEP 1 (ROLE PREDICTION)   We first predict a context-sensitive role *tuple* for each word $w_k$ in the chunk:

$$r^{w_k} = (r_1^{w_k}, r_2^{w_k}, \ldots, r_M^{w_k}) = (a_1^{w_k} \mathbf{r}_1^{chunk}, a_2^{w_k} \mathbf{r}_2^{chunk}, \ldots, a_M^{w_k} \mathbf{r}_M^{chunk}).$$

$a_i^{w_k}$'s represent the distribution of chunk roles, which are predicted by the same RNN used to predict the next tokens. Specifically, we feed the output vector from RNN through a linear layer and then a softmax layer. $\mathbf{r}_i^{chunk}$'s are chunk-level basis role embeddings, shared by all chunks[3]. Step 1 corresponds to the *role prediction* module in Figure 1(c).

STEP 2 (PROJECTION AND BINDING)   For each of the basis roles, we predict their associated filler embeddings. This is done by projecting the HRR word representation $\tilde{E}(w_k)$ (Equation (4)) into $M$ vectors. These fillers are then bound with their corresponding roles and summed together. Specifically,

$$g_1^{w_k}, g_2^{w_k}, \ldots, g_M^{w_k} = [W_1; W_2; \ldots; W_M]^\top \tilde{E}(w_k),$$

$$\hat{E}^c(w_k) = \sum_{i=1}^{M} r_i^{w_k} \circledast g_i^{w_k} = \sum_{i=1}^{M} (a_i^{w_k} \mathbf{r}_i^{chunk}) \circledast g_i^{w_k}.$$

Intuitively, the first two steps aims to evaluate the representational contribution of $w_k$ to the entire chunk $c$. Following the practice in constructing a word-level model (Equation (4)), we also require the separation of different role-filler bindings. We also note that the binding $\hat{E}^c(w_k)$ embeds the word $w_k$ into a space that is specific to the chunk $c$. The first two steps are illustrated in Figure 1(c).

STEP 3 (ENCODING)   After obtaining bindings for all words within a chunk, the chunk embedding is defined as $\hat{E}(c) = \sum_{k=1}^{m} \hat{E}^c(w_k)$ (Figure 1(d)). It can be easily verified that

$$\hat{E}(c) = \sum_{i=1}^{M} \mathbf{r}_i^{chunk} \circledast \left[ \sum_{k=1}^{m} a_i^{w_k} W_i^\top \tilde{E}(w_k) \right] = \sum_{i=1}^{M} \mathbf{r}_i^{chunk} \circledast \hat{E}_i(c). \tag{6}$$

Note that $\hat{E}(c)$ has the same form as Equation (4), and $\hat{E}_i(c)$ can be interpreted as the chunk filler embedding for the $i$-th chunk role. However, chunk embeddings are different in two key aspects. First, the filler embeddings for chunks are projected from the word embeddings, instead of being a set of independently trainable parameters. This compositional approach addresses the first issue regarding the vast number of unique chunks. Second, chunk embeddings rely on weights $a_i^w$'s from role prediction, which provide a natural vehicle for carrying contextual information, therefore addressing the second issue of context-sensitivity.

**Chunk prediction**   Comparable to language modeling that uses next word prediction as supervision, our chunk-level HRR model uses next chunk prediction (CP in Figure 1(a)). Specifically in our experiments, we simply concatenate the chunk embeddings from the last two steps, and feed it through a linear layer followed by $\tanh$ activation:

$$\hat{E}(c_\tau) = \tanh \left( [U_1; U_2]^\top [\hat{E}(c_{\tau-1}); \hat{E}(c_{\tau-2})] \right).$$

---

[3]Note that chunk-level bases $\mathbf{r}_i^{chunk}$'s are different parameters from word-level bases $\mathbf{r}_i^{word}$'s

**Decoding**    Similar to word-level HRR (Equation (5)), we decode the filler embeddings from the predicted chunk embedding using $\mathbf{r}_i^{chunk}$ as cue, and then use the decoded embeddings to factorize the scoring function. The same form of loss function is used. To provide negative examples in the denominator of the softmax Equation (2), we use all the chunks in the mini-batch. These chunks form a pseudo "chunk vocabulary" that is constructed on the fly. Role annealing, and regularization on basis embeddings are also applied.

**Chunk boundaries**    Chunk boundaries have to be supplied in order to construct a chunk embedding. We reply on a third-party chunker to provide such annotations. This is analogous to our word-level model, only that word boundaries are trivially provided by whitespace in English.

## 4    EXPERIMENTS

### 4.1    SETUP

**Data Sets**    We train all models on both Penn Treebank (PTB) (Marcus et al., 1994) and One-Billion-Word Benchmark dataset (1B) (Chelba et al., 2014). For 1B, we experimented with three training sizes: the full dataset, one-tenth of the dataset, and one-hundredth. We evaluate all models on the following three aspects:

- Perplexity: We report perplexity to evaluate language models.
- Intrinsic evaluation: We use a test suite consisting of 18 word embedding benchmark datasets to intrinsically evaluate the quality of word embeddings (Jastrzebski et al., 2017).[4]
- Extrinsic evaluation: We evaluate all word embeddings on six downstream tasks following Nayak et al. (2016): Part-of-speech tagging (POS), chunking, named entity recognition (NER), sentiment analysis (SA), question classification (QC), and natural language Inference (NLI).[5]

**Training details**    For our models, We mainly experimented with two word-level and two chunk-level roles. The basis embeddings are either trained with isometric constraint, or fixed as constant after random initialization.

We also note that for PTB, we experimented with either contiguous input or noncontiguous input. For contiguous input, we follow the common practice in the literature to feed the input sentences in the order that they appear in the document, and initialize the hidden state of LSTM with the last state from the last batch. We refer interested readers to the appendix for more training details.

### 4.2    PERPLEXITY RESULTS

Although not a prime motivation of our approach, Table 1 shows that our HRR models can outperform the baseline in terms of perplexity, especially when we do not have enough training data. On PTB, the best HRR model obtains a gain of 2.2 with noncontiguous input (Fixed-big 88.0 vs Baseline 90.2), and 1.9 with contiguous input (Fixed-big-cont 74.7 vs Baseline-cont 76.6). We also note that increasing the number of basis fillers has a positive impact on HRR models (Fixed-big 88.0 vs Fixed-medium 90.2 vs Fixed-small 97.0). On 1B, we observe that HRR models benefit from having trainable bases with isometric regularization (107.2 vs 112.3 on 1/100 1B, and 55.9 vs 57.9 on full 1B).

### 4.3    WORD LEVEL ANALYSIS

We then demonstrate that our word-level HRRLM can effectively separate certain attributes of word representation. Specifically, we look at our word-level models with two word-level roles, and find that the first set of filler embeddings captures mostly syntax-related categories, especially tense and agreement for verbs, whereas the second set focuses more on the semantic content of words. This decomposition is illustrated in Figure 2, where we visualize both sets of filler embeddings for the

---

[4]`https://github.com/kudkudak/word-embeddings-benchmarks.`
[5]`https://github.com/NehaNayak/veceval`

| Model | PTB | 1/100 1B | 1/10 1B | 1B |
|---|---|---|---|---|
| Baseline | 90.2 | 120.4 | 64.5 | 56.0 |
| Baseline-cont | 76.6 | - | - | - |
| Fixed-small | 97.0 | - | - | - |
| Fixed-medium | 90.2 | - | - | - |
| Fixed-big | **88.0** | 112.3 | 64.2 | 57.9 |
| Fixed-big-cont | **74.7** | - | - | - |
| Isometric-big | 88.2 | **107.2** | **63.9** | **55.9** |

Table 1: Perplexity results on PTB and 1B data sets. All HRR models have two word-level roles, and each role has 320, 200, and 50 basis filler embeddings for Fixed-big, Fixed-medium and Fixed-small, respectively.

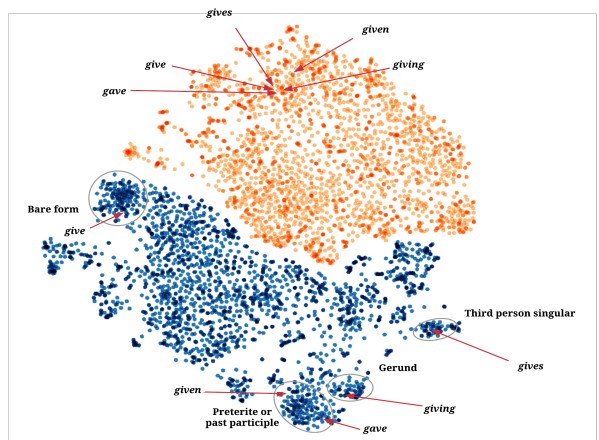

Figure 2: t-SNE visualizations for two sets of filler embeddings (blue and orange).

most frequent 2500 words in PTB via t-SNE (Maaten & Hinton, 2008). In the first set (in blue), verbs with different inflectional markers form distinct clusters. For instance, bare forms, gerunds, preterites, and third-person-singular verbs each form a visually distinguishable group. In contrast, for the second set of filler embeddings (in orange), semantically related words tend to be close regardless of their morphosyntactic markers. For instance, *give*, *gave*, and *giving* are close in the second space.

We proceed to evaluate the quality of the learned word embeddings both intrinsically and extrinsically. As Table 2 summarizes, our HRR models consistently outperform the baseline on the intrinsic evaluation task (numbers on the *left*). The best HRR model Isometric-big obtains a gain of 9.0%, 11.6%, and 16.2% on 1/100 1B, 1/10 1B and full 1B, respectively. In addition, the HRR model with trainable bases has a noticeable advantage over the model with fixed bases (2.5% on average). On the other hand, the HRR models have only a marginal advantage over the baseline (numbers on the *right*), which echos findings in (Schnabel et al., 2015) which claims that intrinsic improvement for word embeddings does not necessarily correlate with improvement in downstream tasks. Detailed results for each dataset and downstream task are available in Table 7 in Appendix B.

| Model | 1/100 1B | 1/10 1B | 1B |
|---|---|---|---|
| Baseline | 0.386\|0.818 | 0.454\|0.816 | 0.427\|0.817 |
| Fixed-big | 0.466\|0.823 | 0.552\|0.817 | 0.543\|0.821 |
| Isometric-big | **0.476\|0.820** | **0.570\|0.819** | **0.589\|0.822** |

Table 2: Averaged scores for intrinsic evaluation (18 datasets, numbers on the left) and extrinsic evaluation (6 tasks, numbers on the right).

| | Overall | Semantics | Syntax | Past tense | Present participle | Plural verbs |
|---|---|---|---|---|---|---|
| Baseline | 0.312 | 0.243 | 0.367 | 0.160 | 0.224 | 0.553 |
| Isometric-big | **0.660** | **0.569** | **0.733** | **0.779** | **0.804** | **0.897** |

Table 3: Accuracy (top 1) on the Semantic-Syntactic analogy benchmark. The most frequent 200K words are used. We use COSMUL for all analogy tests Levy & Goldberg (2014)

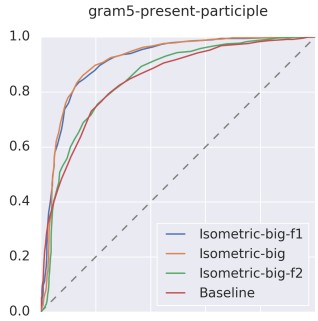
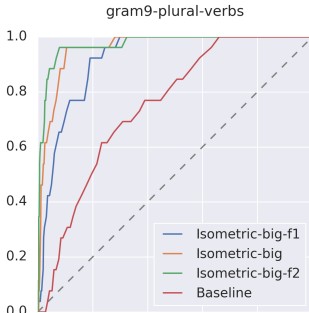

Figure 3: Examples of ROC curves for two categories. The left figure is for a syntax-driven categorization of gerunds, while the right is a semantics-driven categorization of plural verbs. Isometric-big-f1 means the first filler embedding, and Isometric-big-f2 the second.

A breakdown of the intrinsic evaluation reveals that we obtain huge gains on the word analogy task, especially in the verb-related categories such as plural verbs and past tense. On the Semantic-Syntactic analogy dataset Mikolov et al. (2013), the accuracy (top 1) for plural verbs is 0.897 if we consider the most frequent 200K words. If we limit the vocabulary to the more common words (10K), the accuracy for present participles reaches *0.964*. This is consistent with our previous observation that the first set of filler embeddings effectively captures the representation of different verb forms.

To further quantify how decomposed are our word-level filler embeddings, we use them to classify whether two words are related in terms of either semantics or syntax. Specifically, we use a threshold $\theta$ for cosine similarity to determine whether any pair of words $(w, w')$ are related. We then use different values of $\theta$ to plot ROC curve and compute its AUC. For the ground truths, we use the syntactic categories from the Semantic-Syntactic dataset, and extract them from each $a : b = c : d$ example. Two sets of categorizations are obtained, driven by semantics and syntax respectively. For instance, for the example *make : making = give : giving*, we obtain $(\{making, giving\}, \{make, give\})$ for syntactic considerations, and $(\{making, make\}, \{giving, give\})$ for semantics. For the verb-related categories, Table 4 shows that the first set of filler embeddings does noticeably better than the second in the first experiment, while the second set is much better at the second experiment. Figure 3 shows two such categories which confirm that the decomposition does make sense on a crude syntax-semantics level. More details of the intrinsic evaluation results can be found in Table 8 in Appendix B.

| Categorization | Baseline | Filler 1 | Filler 2 |
|---|---|---|---|
| syntactic | 0.842 | **0.912** | 0.840 |
| semantic | 0.670 | 0.873 | **0.975** |

Table 4: Average AUC on verb-related categories for the baseline, and also the two sets of filler embeddings from Isometric-big. All models are trained on the full 1B dataset. The first row corresponds to the first syntactic-driven categorization experiment, and the second row to the semantic-driven experiment.

| Cluster id | Role | Cluster Size | Percentage |
|---|---|---|---|
| 1 | Object | 10 | 80.0% |
| 2 | Begin of sentence | 27 | 100.0% |
| 3 | Prepositional object | 25 | 75.0% |
| 4 | Subject | 13 | 84.6% |
| 5 | Subject | 26 | 88.5% |

Table 5: Cluster analysis for a random selection of 100 sentences containing *the company*. The most dominant role in each cluster is identified by a human judge.

| Metric | Filler 1 | Filler 2 |
|---|---|---|
| Hit rate@1 | 0.604 | 0.572 |
| Hit rate@5 | 0.604 | 0.553 |
| Hit rate@10 | 0.600 | 0.544 |
| Hit rate@25 | 0.586 | 0.531 |
| Purity | 0.677 | 0.610 |

Table 6: Hit rate and purity score analysis for chunk embeddings. See Sec.4.4 for more details.

### 4.4 CHUNK LEVEL ANALYSIS

We train a chunk-level HRR model by initializing it with a pretrained word-level model on PTB.[6] We evaluate the quality of chunk embeddings in two ways. First, we perform a human analysis focused on the phrase *the company*, which is the most frequent noun phrase in PTB. We randomly select 100 occurrences, and cluster their chunk embeddings using $K$-means into 5 categories based on the chunk-level filler embeddings.[7] For each sentence cluster we manually identify the dominating role which *the company* played in that cluster of sentences. Table 5 shows that there is a clear role for the phrases in each of the clusters. We also performed a t-SNE analysis using the second filler embedding, in which we observed *the company* is close to many semantically related nouns such as *stock*, *market* and etc. From this analysis we see that the first role in chunk-level HRR is more sensitive to the different syntactic roles *the company* plays dependent on the context, while the second role is more associated with its related semantic concepts.

Secondly, to further investigate if our model consistently captures chunk-level roles, we run the model on the Wall Street Journal part of the OntoNotes dataset (Weischedel et al., 2013) in which semantic role labels for each sentence are provided, and compare chunk embedding clusters against the role labels. We are mainly interested in 4 coarser-grained labels corresponding to AGENT, PATIENT, PREDICATE and MODIFIER. We provide two metrics to measure the quality of the obtained clusters. First, for each chunk $c$, we find its $k$ nearest neighbors ($k = 1, 5, 10, 25$) according to their filler embeddings, and report the percentage of the cases where $c$ shares the same role labels as its neighbor. This hit-rate metric gives us a sense of how accurately chunk clusters capture meaningful semantic roles. In addition, we compute purity score for the learned clusters against the ground truth clusters. Both metric scores are reported in Table 6, from which we can see that chunk embedding indeed correctly captures semantic roles more than half of the time. Moreover, the first set of fillers are more accurate at clustering these roles than the second, and also deteriorates better when we include more neighbors to compute the hit rates.

## 5 RELATED WORK

Perhaps most related to our work are recent attempts to integrate tensor product structure with neural networks (Palangi et al., 2017; Huang et al., 2018). While these works and ours share a common goal of incorporating neural models with symbolic structures, there are several difference. First of all, we make use of HRR instead of tensor product as basis for our representation to enable long

---

[6]The perplexity for the chunk-level model is 89.5 vs 88.0 for the word-level model. The degradation of performance for next word prediction is noticeable. However, we do note that such degradation is relatively common for some multitask settings.

[7]More details about the experimental setup for chunk level analysis are provided in Appendix C.

sequence encoding without parameter explosion. Moreover, we aim to induce linguistic structures from a task that requires as little supervision as possible like language modeling, whereas their work is focused on question answering which provides stronger guidance signal from labeled data. On the other hand, recent work (Shen et al., 2018) also proposes a novel network architecture that is capable of learning syntactic roles and semantics jointly, but it is not based on structured representation like HRR.

There have been many attempts of using symbolic architectures like HRR and TPR for linguistic analysis, see for example (Jones & Mewhort, 2007; De Vine & Bruza, 2010; Recchia et al., 2015; Prince & Smolensky, 1997; Clark et al., 2008; Clark & Pulman, 2010; Grefenstette et al., 2011). HRR itself as a variable binding and association mechanism, has also been integrated with neural networks with different motivations like associative memory modeling (Danihelka et al., 2016), relationship reasoning (Weiss et al., 2016) etc. While most of the work mainly focuses on symbolic and formal analysis via algebraic operations and logic derivations, our work aims to enable neural language models to learn linguistic roles by taking advantage of the HRR properties.

## 6   CONCLUSION

In this paper, we employ HRR to provide a principled decomposition of representation. We design our HRR language models to work on both word-level and chunk-level. Our analysis revealed that by introducing an inductive bias, our models can learn disentangled representations, which roughly corresponds to syntax and semantics.

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

## APPENDIX A    TRAINING DETAILS

We use the same architecture for all models on both LM data sets, with a two-layered LSTM (Hochreiter & Schmidhuber (1997)) and tied input and output embeddings. Unless otherwise noted, the dimensionality of word embeddings and hidden states for LSTM are all 650 and uniformly initialized in (-0.05, 0.05). For PTB with contiguous input, we follow the existing literature to use a batch size of 20, backproprogate for 35 steps (Zaremba et al., 2014), and use SGD with an initial learning rate of 1.0. A decay of 0.8 is applied whenever the perplexity validation set does not improve. We use ADAM (Kingma & Ba (2014)) with an initial learning rate $8e^{-5}$ to train all models for the One-Billion-Word benchmark.

We used a third-party chunker to provide chunk boundaries for the entire PTB data set (Daelemans & Van den Bosch, 2005).[8]

## APPENDIX B    INTRINSIC AND EXTRINSIC EVALUATION FOR WORD EMBEDDINGS

We present detailed results for each of the 18 word embedding dataset in Table 7. They can be divided into three types of experiments: categorization, similarity, and analogy. Average scores for each type and also overall average are provided. Results for three models (Baseline, Isometric-big, and Fixed-big) are presented on 1B with different training sizes.

For the Semantic-Syntactic analogy dataset (Google), we also provide a detailed breakdown of each test category in Table 8. The HRR model scores very high for many syntactic categories especially related to verbs.

---

[8] https://www.clips.uantwerpen.be/pages/MBSP

| | 1/100 1B | | | 1/10 1B | | | 1B | | |
|---|---|---|---|---|---|---|---|---|---|
| | Base | HRR | Fixed-HRR | Base | HRR | Fixed-HRR | Base | HRR | Fixed-HRR |
| AP | 0.476 | 0.610 | 0.595 | 0.592 | 0.626 | 0.642 | 0.601 | 0.664 | 0.672 |
| BLESS | 0.462 | 0.527 | 0.538 | 0.635 | 0.730 | 0.715 | 0.710 | 0.785 | 0.730 |
| Battig | 0.247 | 0.334 | 0.327 | 0.315 | 0.408 | 0.425 | 0.350 | 0.470 | 0.446 |
| ESSLI_1a | 0.523 | 0.659 | 0.659 | 0.705 | 0.750 | 0.795 | 0.727 | 0.773 | 0.727 |
| ESSLI_2b | 0.775 | 0.775 | 0.800 | 0.800 | 0.900 | 0.900 | 0.800 | 0.800 | 0.750 |
| ESSLI_2c | 0.444 | 0.600 | 0.533 | 0.533 | 0.667 | 0.644 | 0.533 | 0.578 | 0.667 |
| MEN | 0.408 | 0.513 | 0.481 | 0.475 | 0.619 | 0.570 | 0.328 | 0.604 | 0.487 |
| MTurk | 0.485 | 0.529 | 0.524 | 0.459 | 0.609 | 0.544 | 0.234 | 0.582 | 0.476 |
| RG65 | 0.356 | 0.424 | 0.428 | 0.434 | 0.462 | 0.474 | 0.414 | 0.575 | 0.554 |
| RW | 0.330 | 0.431 | 0.430 | 0.373 | 0.461 | 0.435 | 0.337 | 0.468 | 0.412 |
| SimLex999 | 0.221 | 0.243 | 0.237 | 0.340 | 0.382 | 0.376 | 0.331 | 0.416 | 0.382 |
| TR9856 | 0.406 | 0.447 | 0.454 | 0.445 | 0.544 | 0.511 | 0.339 | 0.555 | 0.476 |
| WS353 | 0.477 | 0.519 | 0.509 | 0.483 | 0.592 | 0.540 | 0.414 | 0.586 | 0.523 |
| WS353R | 0.363 | 0.412 | 0.406 | 0.342 | 0.501 | 0.419 | 0.232 | 0.482 | 0.371 |
| WS353S | 0.618 | 0.679 | 0.660 | 0.607 | 0.698 | 0.670 | 0.567 | 0.697 | 0.664 |
| Google | 0.057 | 0.203 | 0.180 | 0.227 | 0.515 | 0.482 | 0.312 | 0.660 | 0.590 |
| MSR | 0.105 | 0.438 | 0.413 | 0.234 | 0.627 | 0.605 | 0.285 | 0.728 | 0.663 |
| SemEval2012_2 | 0.169 | 0.180 | 0.170 | 0.193 | 0.217 | 0.218 | 0.215 | 0.228 | 0.217 |
| categorization | 0.488 | 0.584 | 0.575 | 0.597 | 0.680 | 0.687 | 0.620 | 0.678 | 0.665 |
| similarity | 0.408 | 0.466 | 0.459 | 0.440 | 0.541 | 0.504 | 0.355 | 0.553 | 0.484 |
| analogy | 0.110 | 0.273 | 0.255 | 0.218 | 0.453 | 0.435 | 0.271 | 0.539 | 0.480 |
| overall | 0.385 | 0.473 | 0.464 | 0.455 | 0.573 | 0.554 | 0.429 | 0.592 | 0.545 |

Table 7: Word analogy test results on different datasets.

| Category | Isometric-big | | | Baseline | | |
|---|---|---|---|---|---|---|
| | Correct | Total | Accuracy | Correct | Total | Accuracy |
| comparative | 1130 | 1332 | 0.848 | 839 | 1332 | 0.630 |
| plural | 1154 | 1332 | 0.866 | 900 | 1332 | 0.676 |
| superlative | 848 | 1122 | 0.756 | 321 | 1122 | 0.286 |
| plural-verbs | 780 | 870 | 0.897 | 481 | 870 | 0.553 |
| opposite | 284 | 812 | 0.350 | 167 | 812 | 0.206 |
| nationality-adjective | 1255 | 1521 | 0.825 | 618 | 1521 | 0.406 |
| past-tense | 1215 | 1560 | 0.779 | 249 | 1560 | 0.160 |
| present-participle | 849 | 1056 | 0.804 | 237 | 1056 | 0.224 |
| adjective-to-adverb | 254 | 992 | 0.256 | 71 | 992 | 0.072 |
| currency | 171 | 646 | 0.265 | 95 | 646 | 0.147 |
| capital-world | 3176 | 4291 | 0.740 | 1284 | 4291 | 0.299 |
| capital-common-countries | 445 | 506 | 0.879 | 159 | 506 | 0.314 |
| family | 417 | 506 | 0.824 | 369 | 506 | 0.729 |
| city-in-state | 578 | 2467 | 0.234 | 137 | 2467 | 0.056 |

Table 8: Detailed word analogy test results in each category.

## APPENDIX C   CHUNK LEVEL ANALYSIS

To obtain clustering results, we used `sklearn` with the default parameters and settings provided by the toolbox. For the OntoNotes experiment, we removed sentences that have unknown tokens of more than 10 percent, and removed chunks that are longer than six words. In addition, we removed some rare role labels (out of 11), and collapsed some object-related roles (e.g., direct object, and indirect object) into PATIENT. This resulted in a total of around 4K chunks, and four remaining coarse labels AGENT, PATIENT, PREDICATE, and MODIFIER, which correspond to the original tags ARG0, ARG1, V and ARGM.

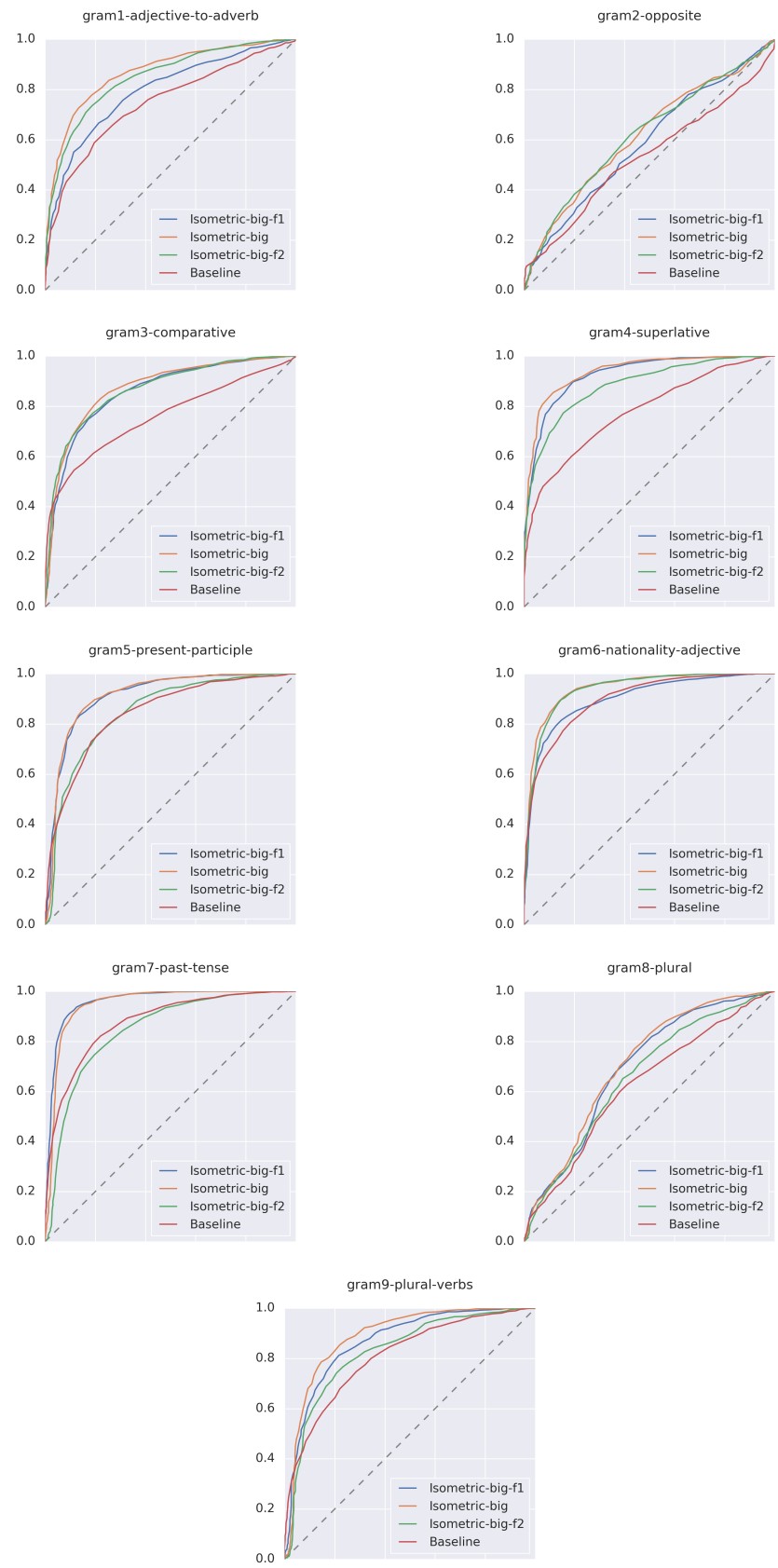

Figure 4: ROC for every syntactic category in the first set of categorization experiments (syntax-driven).

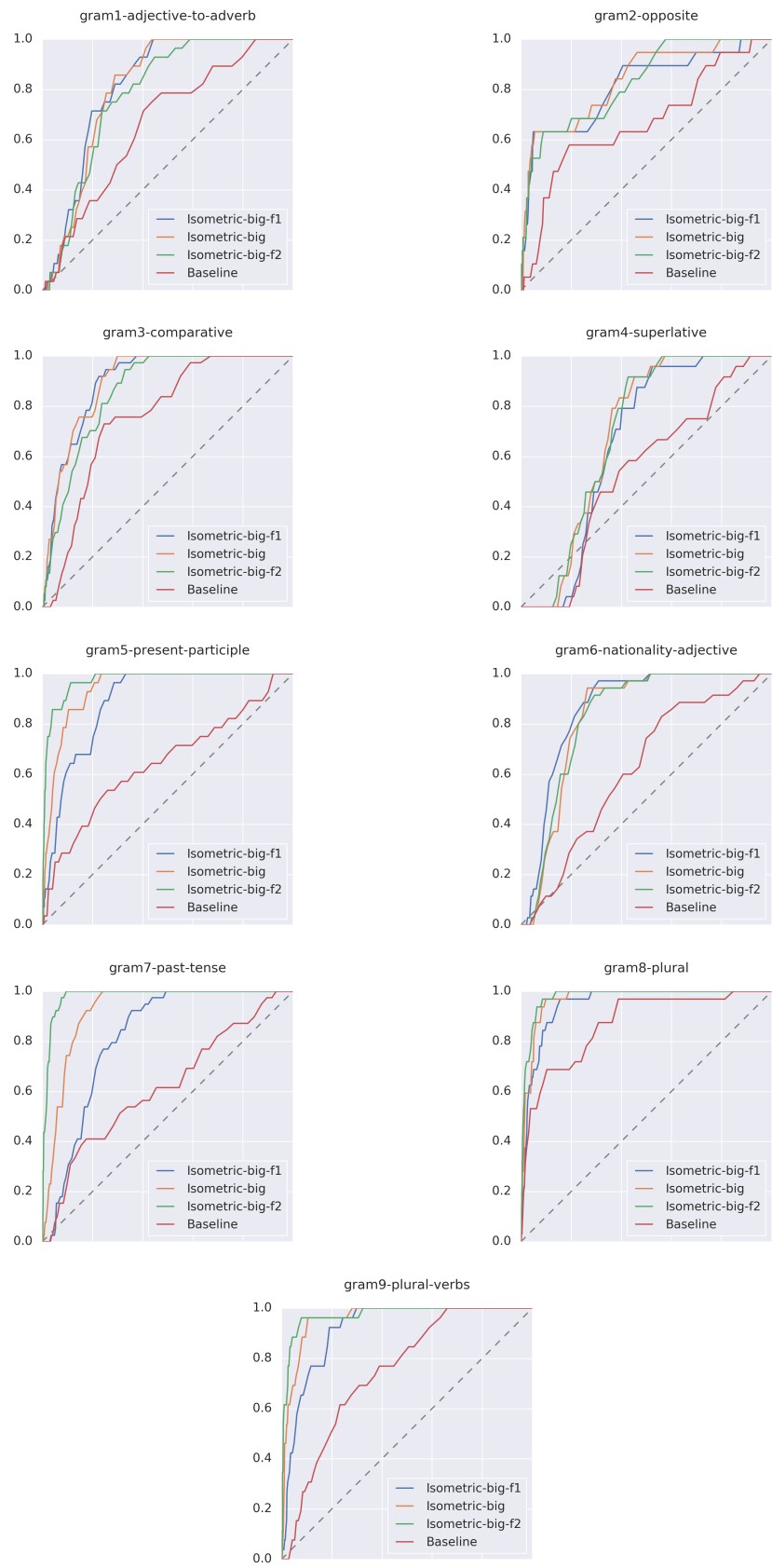

Figure 5: ROC for every syntactic category in the second set of categorization experiments (semantics-driven).

