# OpenReview forum: "Towards Decomposed Linguistic Representation with Holographic Reduced Representation"
_ICLR.cc/2019/Conference_

### Official Review · AnonReviewer3 · 2018-11-02
**Novel approach to learning decomposable representations; some unclear parts and questionable validity; weak performance**

**Rating:** 6
**Confidence:** 3

**Review:**


Summary:
========
Theis paper proposes a method for learning decomposable representations in the context of a language modeling task. Using holographic reduced representations (HRR), a word embedding is composed of a role and a filler. The embedding is then fed to an LSTM language model. There is also an extension to chunk-level representations. Experimentally, the model achieves perplexity comparable to a (weak) baseline LSTM model. The analysis of the learned representations shows a separation into syntactic and semantic roles.

The paper targets an important problem, that of learning decomposable representations. As far as I know, it introduces a novel perspective using HRR and does so in the context of language modeling, which is a core NLP task. The analysis of the learned representations is quite interesting. I do have some concerns with regards to the quality of the language model, the clarity of some of the model description, and the validity of using HRR in this scenario. Please see detailed comments below.

Comments:
=========
1. Section 2 refers to Plate (1995) for the conditions when the approximate decoding via correlation holds. I think it's important to mention these conditions and discuss whether they apply to the language modeling case. In particular, Plate mentions that the elements of each vector need to be iid with mean zero and variance 1/n (where n is the length of the vector). Is this true for the present case? Typically, word embeddings and LSTM states are do not exhibit this distribution. Are there other conditions that are (not) met?
2. Learning separate bases for different role-filler bindings is said to encourage the model to learn a decomposition of word representation. On the other hand, if I understand correctly, this means that word embeddings are not shared between roles, because s^w_i is also a role-specific vector (not just a word-specific vector). Is that a cause of concern?
3. It's not clear to me where in the overall model the next word is predicted. Figure 1b has an LSTM that predicts filler embeddings. Does this replace predicting the next word in a vanilla LSTM? Equation 5 still computes a word score. Is this used to compute the probability of the next word as in equation 2?
4. Comparison to other methods for composing words. Since much of the paper is concerned with composing words, it seem natural to compare the methods (and maybe some of the results) to methods for composing words. Some examples include [2] and the line of work on recursive neural networks by Socher et al., but there are many others.
5. Perplexity results:
- The baseline results (100.5 ppl on PTB) are very weak for an LSTM. There are multiple papers showing that a simple LSTM can do much better. The heavily tuned LSTM of [1] gets 59.6 but even less tuned LSTMs go under 80 or 80 ppl. See some results in [1]. This raises a concern that the improvements from the HRR model may not be significant. Would they hold in a more competitive model?
- Can you speculate or analyze in more detail why the chunk-level model doesn't perform well, and why adding more fillers doesn't help in this case?
6. Motivation:
- The introduction claims that the dominant encoder-decoder paradigm learns "transformations from many smaller comprising units to one complex emedding, and vice versa". This claim should be qualified by the use of attention, where there is not a single complex embedding, rather a distribution over multiple embeddings.
- Introduction, first paragraph, claims that "such crude way of representing the structure is unsatisfactory, due to a lack of transparency, interpretability or transferability" - what do you mean by these concepts and how exactly is the current approach limited with respect to them? Giving a bit more details about this point here or elsewhere in the paper would help motivate the work.
7. Section 3.3 was not so clear to me:
- In step 1, what are these r_i^{chunk}? Should we assume that all chunks have the same role embeddings, despite them potentially being syntactically different? How do you determine where to split output vectors from the RNN to two parts? What is the motivation for doing this?
- In prediction, how do you predict the next chunk embedding? Is there a different loss function for this?
- Please provide more details on decoding, such as the mentioned annealing and regularization.
- Finally, the reliance on a chunker is quite limiting. These may not be always available or of high quality.
8. The analysis in section 4.3 is very interesting and compelling. Figure 2 makes a good point. I would have liked to see more analysis along these lines. For example, more discussion of the word analogy results, including categories where HRR does not do better than the baseline. Also consider other analogy datasets that capture different aspects.
9. While I agree that automatic evaluation at chunk-level is challenging, I think more can be done. For instance, annotations in PTB can be used to automatically assign roles such as those in table 4, or others (there are plenty of annotations on PTB), and then to evaluate clustering along different annotations at a larger scale.
10. The introduction mentions a subset of the one billion word LM dataset (why a subset?), but then the rest of the papers evaluates only on PTB. Is this additional dataset used or not?
11. Introduction, first paragraph, last sentence: "much previous work" - please cite such relevant work on inducing disentangled representations.
12. Please improve the visibility of Figure 1. Some symbols are hard to see when printed.
13. More details on the regularization on basis embeddings (page 4) would be useful.
14. Section 3.3 says that each unique word token is assigned a vectorial parameter. Should this be word type?
15. Why not initialize the hidden state with the last state from the last batch? I understand that this is done to assure that the chunk-level models only consider intra-sentential information, but why is this desired?
16. Have you considered using more than two roles? I wonder how figure 2 would look in this case.


Writing, grammar, etc.:
======================
- End of section 1: Our papers -> Our paper
- Section 2: such approach -> such an approach; HRR use -> HRR uses; three operations -> three operations*:*
- Section 3.1: "the next token w_t" - should this be w_{t+1)?
- Section 3.2, decoding: remain -> remains
- Section 3.3: work token -> word token
- Section 4.1: word analogy task -> a word analogy task; number basis -> numbers of basis
- Section 4.2: that the increasing -> that increasing
- Section 4.3: no space before comma (first paragraph); on word analogy task -> on a word analogy task; belong -> belongs
- Section 4.4: performed similar -> performed a similar; luster -> cluster
- Section 5: these work -> these works/papers/studies; share common goal -> share a common goal; we makes -> we make; has been -> have been

References
==========
[1] Melis et al., On the State of the Art of Evaluation in Neural Language Models
[2] Mitchell and Lapata, Vector-based Models of Semantic Composition

---

> ### Author Response · Authors · 2018-11-08
> **Response to reviewer 3: Part 1**
>
> Thank you for your kind words and very detailed comments.
>
> Before delving into the detail, we would appreciate it if you could elaborate a bit more on the concern about “the validity of using HRR in this scenario” mentioned in the second paragraph? (a) Did you mean using HRR in language modeling? (b) If this is case, does it concern you because the choice of this task, or because of the inadequate baseline performance? Thank you very much!
>
> (1) “the conditions when the approximate decoding via correlation holds”
>
> Thanks for pointing this out. We will explain a bit more in our updated version. For our experiments though, only the models run with fixed basis embeddings are with mean zero and variance 1/n because they are randomly sampled and fixed throughout training. I agree that word embeddings and LSTM states do not typically exhibit such a distribution (especially iid condition). However, we would also like to make two points. First, past work [1] that successfully uses HRR as associative memory, where these conditions are also not explicitly met (or at least they didn’t show it). Second, in our case, the decomposed scoring function actually acts as an (soft) enforcer that makes sure decoding works properly. The loss would only go down when the predicted filler embedding (after decoding) is close to the original filler embedding (before encoding). This is largely mediated by dot product -- the more accurate the decoding is, the bigger the value of dot product is.
>
> As for other conditions, it is also required that the dimensionality of the vector be sufficiently bigger than the number of stored items. This obviously holds in our case since we are only using a couple of variable bindings, and we will make it more clear in the updated version.
>
> (2) “Learning separate bases for different role-filler bindings is said to encourage the model to learn a decomposition of word representation”
>
> If we understand the question correctly, for each word, there are two (equal to the number of roles) filler embeddings, which have separate bases. These filler embeddings are then bound with their associated role embeddings. In this sense, base filler embeddings and role embeddings are shared across all words, but not between roles. Our earlier experiments showed that without separating these bases, decomposition of representations did not occur. We think it makes sense intuitively -- a decomposition of representation usually necessitates a separation of feature space. Does this address your concern?
>
> (3) “It's not clear to me where in the overall model the next word is predicted”
>
> We apologize for this confusion. The decoding module in 1(b) corresponds to equation 5. Instead of using one dot product as in a vanilla LSTM, we use the sum of two dot products, each of which is responsible for one role-filler binding.
>
> Indeed the score in equation 5 is used similarly as in equation 2. We will make this more clear in the updated version.
>
> (4) “Comparison to other methods for composing words”
>
> If we understand it correctly, you are referring to the word-level model since this is where we spent most time entailing and analyzing. As we argued in the response to reviewer 2 (point 4, reproduced below), we do not find any directly comparable method to the best our knowledge.
>
> “...There are certainly many existing methods that try to incorporate structures, but mostly to enhance their representation, not decompose their representation. Moreover, the unsupervised nature of our approach makes direct comparison even harder.”
>
> The cited work you provided [2], and also Socher’s recursive network network deal with composing phrases from individual words, which does not concern the decomposition of word representation. Moreover, recursive neural networks need additional input such as parsed trees, which is definitely outside the scope of our paper.
>
> We would like to emphasize that the main contribution is about the decomposition/separation of representations. This decomposition, in HRR’s framework, is accompanied by the initial operation of encoding/composing. Due to space limit, we do not fully investigate the potential advantage/disadvantage of using HRR as an encoder (compared to (say) Socher’s work), but rather spend most of the time using HRR to set up a model that can induce decomposition.
>
> Of course, we can be totally ignorant of other directly comparable methods. If you have any specific method in mind, we would really appreciate it if you can provide us some pointers.

---

> ### Author Response · Authors · 2018-11-08
> **Response to reviewer 3: Part 2**
>
> (5) (i) “weak baseline results”
>
> This is a very valid concern, which reviewer 2 also raised (point 3). We reproduce our response below.
>
> “We are fully aware that our baseline seems to underperform, as pointed out by reviewer 3 as well. First we would like to point out that contrary to common practice in LM literature, ‘we do not assume that the contiguous sentences in the raw data are fed sequentially as input’, and as a result ‘we do not initialize the hidden state of LSTM with the last state from the last batch’ (section 4.1)...
>
> On the other hand, we are also running another word-level baseline that follows the common practice in LM literature. We will update the results shortly.”
>
> (ii) “Can you speculate or analyze in more detail why the chunk-level model doesn't perform well, and why adding more fillers doesn't help in this case? “
>
> We speculate that it’s because chunk prediction doesn’t provide much complemental information for word prediction, and as a result, a competing/non-beneficial chunk prediction loss doesn’t help bring the word-level loss further down. This could potentially be mitigated by a more powerful model (say, bigger and deeper model), but it might cause more overfitting on PTB. Our preliminary results showed that introducing chunk-level loss at later stage of training helps bring down the perplexity, but we will provide more experimental results to make a conclusive judgment.
>
> (6) (i) “This claim should be qualified by the use of attention”
>
> Thanks for pointing this out. We will say a bit more in a footnote. We would also like to note that the use of attention is accompanied by what is essentially a memory mechanism (multiple embeddings). This is definitely related to using HRR as associated memory.
>
> (ii) “what do you mean by these concepts and how exactly is the current approach limited with respect to them?”
>
> Thanks for pointing it out, and we would add more details in the update version. By transparency and interpretability, we mean that the operations of encoding and decoding have clear conceptual meaning. In our case, it is manifested by the explicit role-filler binding. Transferability means that some features are only transferable in certain aspects. For instance, a separation of domain-specific features from domain-invariant features would help the latter transfer more easily to other related tasks.
>
> (7) “Section 3.3 was not so clear to me:”
>
> Apologies for the confusion. We would make it more clear in the updated version.
>
> (i) r_i^{chunk}s are indeed shared by all words, but because we also have context-sensitive weights (step 1), the associated role for the chunk would be different. As for splitting the output vectors, we meant that we use the same LSTM to predict two vectors -- one for predicting next word, the other for predicting the chunk-specific role weights a’s. Sharing the same RNN hidden state is not necessary, but we found it effective without introducing another neural network.
>
> (ii) As for chunk prediction, it is done by concatenating the previous two chunk embeddings as input, and feed it through a linear layer followed by tanh (page 5, paragraph Prediction). The same form of loss function is used (sum of dot products), but the negative samples (in the denominator) are taken from the same batch (page 5, paragraph Decoding). We will add more details to these two paragraphs.
>
> (iii) As for the chunker, we fully acknowledge its limitation. It will be ideal if chunking is done jointly with LM, but it is outside the scope of this paper. However, using a chunker makes intuitive sense, and is analogous to what we have done to the word-level model. Specifically, the word-level model needs word boundaries, which are naturally provided by whitespaces for languages like English. Similarly, the chunk-level model needs chunk boundaries, which is provided by a chunker.

---

> ### Author Response · Authors · 2018-11-08
> **Response to reviewer 3: Part 3**
>
> (8) “The analysis in section 4.3”
>
> Thank you for the kind words. We will add more categories. We are also considering adding more datasets, and possibly adding more analysis in the appendix.
>
> (9) “automatic evaluation at chunk-level is challenging”
>
> We initially refrained from extracting roles from PTB because we couldn’t find any existing script to do that. Of course, extracting phrases is easy but to our best knowledge not for roles. One way is to use dependency relations. But those are usually very diverse and nuanced, and we had concerns about how well our unsupervised method would fare. Another way is to use semantic role labeling, even though a similar concern arises. We are currently expanding our experiment section to these two tasks , which also address your concern here.
>
> (10) one-billion-word dataset
>
> We reproduce our response to a relevant point raised by reviewer 2 below.
>
> “...we are currently expanding our experiments to more datasets (wiki, one-billion-word, possibly some domain-specific texts, or some subset of them)”
>
>
> (11-14)
>
> Thanks for the comments. We would address these detailed issued in the update version.
>
> (15) “Why not initialize the hidden state with the last state from the last batch”
>
> We reproduce our response to a relevant point raised by reviewer 2 below.
>
> “...We took this approach to ensure that chunk-level representations capture only intra-sentential roles -- we do not consider discourse-level features. The downside of this is that we can no longer reply on information from the last sentence to help predict the current one.”
>
> (16) “Have you considered using more than two roles?”
>
> Yes indeed. However, we did not observe further decomposition on PTB. We suspect that there are two possibilities that need more consideration. First, the model simply needs more data to achieve decomposition into even more aspects. Second, the signal from LM might not be strong enough to induce even more separated aspects. We think that the second issue is outside the scope of the current submission, and as for the first issue, we are currently running experiments on a bigger scale, and will update our results shortly.
>
> (17) “writing, grammar”
>
> Noted. Thanks for pointing them out.

---

> ### Comment · AnonReviewer3 · 2018-11-18
> **Thank you for your detailed response; some additional comments**
>
> Thank you for your very detailed response. I would be happy to read an updated version that takes into account the comments by the reviewers and reconsider my evaluation accordingly.
>
> Below are some additional comments.
>
> 1. My concern about the validity of using HRR was mainly referring to the conditions when decoding works in HRR, which you better explained in your response. I understand that the method might somehow work even though the distributional conditions do not hold, but I still do not understand what the implications are. It seems like a crucial assumption. Is there any way to evaluate or estimate the effect of the conditions not holding in your case?
>
> 2. Separate bases for different filler embeddings: yes, what you say makes sense, as separating the feature spaces may be required for decomposing the representation. I can see that in clear-cut cases (e.g., two separate meanings of a word like "bank"). But might there be cases where it may be worth sharing information between roles?
>
> 4. Decomposing representations versus composing words: thank you for clarifying this point. It seems like a confusion on my part, but perhaps a note on this point might help the confused reader.
>
> 5. Weak baseline results:
> (i) I look forward to seeing the updated results with stronger baselines. On the matter of initializing the hidden state from the last batch, I'm not sure that would make a big difference in practice, but you might as well try that too. Regarding point (15), I don't see a reason to limit to intra-sentential roles, to the extent that this initialization makes a difference.
>
> (ii) On the speculation that "chunk prediction doesn’t provide much complemental information for word prediction" - could you test that by looking at specific examples where one method works better than the others?
>
> 9. Both dependency relations and semantic roles (or semantic dependencies: http://sdp.delph-in.net) would be very interesting in my opinion. You can look at the major relations or coarser categorizations if you're concerned with their diversity.
>
> 16. It is rather disappointing that no additional decomposition is obtained with more than two roles. Can you provide more details? Are some roles not used at all or are some used for the same function? My guess is that PTB should have enough data for further decomposition, but it would be interesting to see if more decomposition emerges in a larger dataset.

---

> > ### Author Response · Authors · 2018-12-14
> > **Response to your additional comments**
> >
> > Thank you again for the very careful and detailed review. We answer your questions below:
> >
> > 1. We agree that the distributional conditions are crucial to the success of decoding procedure. One way of evaluating the sensitivity of model performance to the degree of satisfying these conditions is that for our models with fixed basis, we draw basis embeddings from arbitrary distributions, with different means and variances, then compare the difference in model performance. Drawing embedding vectors from an arbitrary distribution with very high variance violates the HRR condition, and we expect the model performance to drop sharply in these settings.
> >
> > 2. Yes, you are right that information can be shared between different roles, and our formulation of the model makes simplified assumptions about many subtle linguistic phenomena. We formulated the model with separate bases for different roles out of the motivation of easing the training procedure, that is to facilitate the model to learn disentanglement. Using a set of shared bases brings difficulty into training that might lead to “role collapse”, in which case a single role explains everything. An ideal case would be a middle ground between the two extremes: we provide the model a set of shared bases, and the model learns to select which bases to use for each role. That would probably require adding another latent variable conditioned on the role, and is certainly something interesting for us to explore in our future work.
> >
> > 4. Thanks for your suggestion. We have given further clarification on this point in our revised submission.
> >
> > 5. (i) We re-trained our models and improved the baseline results in our revised submission. Our implementation is based upon the open-source Tensorflow RNNLM implementation, and the results are comparable to the scores reported in the document. In our experiments, constraining roles to be intra-sentential simplifies evaluation on the chunk level, since most annotations are provided in a sentence by sentence basis.
> >
> > (ii). Our chunk-level model is trained with both a word prediction loss term and a chunk prediction loss term, and is fine-tuned on a pre-trained word-level model. The fact that the perplexity is slightly worse after fine-tuning suggests that chunk prediction might not provide much information for word prediction. Looking at some specific examples is a bit problematic since it might lead to overgeneralization if we only examine a few samples. A more systematic analysis is warranted.
> >
> > 9. Thank you for your suggestion. Our revised submission includes an experiment which compares the predicted chunk-level roles and coarser-grained semantic roles labels provided by the OntoNote dataset. The results suggests that indeed the predicted chunk-level roles correspond with the ground-truth semantic roles most of the time.
> >
> > 16. Our full 1B experiment with 4 roles shows that the separation of roles becomes more “gradual” in contrast to a two-role model. Specifically, whereas a two-role model has a clear contrast between the first role and the second, a four-role model has increasing sensitivity to semantics from the first role to the fourth, and decreasing sensitivity to verb forms. For instance, the average intra-word cosine similarity for ‘getting’, ‘get’, ‘gets’ and ‘got’ is 0.286, 0.444, 0.508, 0.510 for the first role to the four role. Although we believe that the corpus contains a myriad of contextual information for better separation of roles, we think a more targeted loss function might be needed, perhaps along the line of [1].
> >
> > [1] Assessing the Ability of LSTMs to Learn Syntax-Sensitive Dependencies. Tal Linzen,  Emmanuel Dupoux and Yoav Goldberg. TACL 2016.

---

### Official Review · AnonReviewer1 · 2018-11-03
**Back to the past**

**Rating:** 5
**Confidence:** 4

**Review:**

This paper is very interesting as it seems to bring the clock back to Holographic Reduced Representations (HRRs) and their role in Deep Learning. It is an important paper as it is always important to learn from the past. HRRs have been introduced as a form of representation that is invertible. There are two important aspects of this compositional representation: base vectors are generally drawn from a multivariate gaussian distribution and the vector composition operation is the circular convolution. In this paper, it is not clear why random vectors have not been used. It seems that everything is based on the fact that orthonormality is impose with a regularization function. But, how can this regularization function can preserve the properties of the vectors such that when these vectors are composed the properties are preserved.

Moreover, the sentence "this is computationally infeasible due to the vast number of unique chunks" is not completely true as HRR have been used to represent trees in "Distributed Tree Kernels" by modifying the composition operation in a shuffled circular convolution.

---

> ### Author Response · Authors · 2018-11-08
> **Response to reviewer 1**
>
> Thank you for your interest in our work and your kind words regarding the direction our paper takes. We summarize all the concerns raised and provide a point-by-point response below.
>
> (1) “In this paper, it is not clear why random vectors have not been used”
>
> We have two points to make regarding this comment. First, we did experiment on using fixed random basis embeddings, be it basis role embeddings or basis filler embeddings. This is denoted by models with names Fixed-* in Table 1. This is also mentioned in Page 4, right below Figure 1, “we also consider using fixed random vectors for basis embeddings”. Second, if you are referring to using random vectors for not just bases, but also other trainable word-embedding related parameters (such as s^w_i), we think it is better to treat them as learnable parameters since random vectors do not cluster together in a meaningful way that corresponds to natural language.
>
> (2) “But, how can this regularization function preserve the properties of the vectors such that when these vectors are composed the properties are preserved”
>
> We agree with this characterization. However, we want to remake the point we make in response to reviewer 3 (point 1, reproduced below)
>
> “...in our case, the decomposed scoring function actually acts as an (soft) enforcer that makes sure decoding works properly. The loss would only go down when the predicted filler embedding (after decoding) is close to the original filler embedding (before encoding). This is largely mediated by dot product -- the more accurate the decoding is, the bigger the value of dot product is.”
>
> Although it is out our intention to design a theoretically complete model that preserves the properties all the way through, we do mean to take advantage of HRR properties, combined with black-box modeling from neural networks. We believe this is a reasonable approach to take in order to make our model viable in the world of deep learning.
>
> (3) “Moreover, the sentence ‘this is computationally infeasible due to the vast number of unique chunks’ is not completely true”
>
> We meant to say that directly extending our word-level model to chunk-level is not plausible, because for word-level model, we designate a learnable vectorial parameter to each word type. By analogy, we would have to use a learnable vectorial parameter for each unique chunk type, which renders it intractable in our case. It is in this sense that we meant by saying “this is computationally infeasible”.
>
> Hopefully these responses address your concerns.

---

### Official Review · AnonReviewer2 · 2018-11-06
**Decomposed Linguistic Representation with Holographic Reduced Representations**

**Rating:** 5
**Confidence:** 4

**Review:**

The paper proposes a new approach for neural language models based on holographic reduced representations (HRRs). The goal of the approach is to learn disentangled representations that separate different aspects of a term, such as its semantic and its syntax. For this purpose the paper proposes models both on the word and chunk level. These models aim disentangle the latent space by structuring the latent space into different aspects via role-filler bindings.

Learning disentangled representations is a promising research direction that fits well into ICLR. The paper proposes interesting ideas to achieve this goal
in neural language models via HRRs. Compositional models like HRRs make a lot of sense for disentangling structure in the embedding space. Some of the experimental results seem to indicate that the proposed approach is indeed capable to discover rough linguistic roles. However, I am currently concerned about different aspects of the paper:

- From a modeling perspective, the paper seems to conflate two points: a) language modeling vie role-filler/variable-binding models and b) holographic models as specific instance of variable bindings. The benefits of HRRs (compared e.g., to tensor-product based models) are likely in terms of parameter efficiency. However, the benefits from a variable-binding approach for disentanglement should remain across the different binding operators. It would be good to separate these aspects and also evaluate other binding operators like tensors products in the experiments.

- It is also not clear to me in what way we can interpret the different filler embeddings. The paper seems to argue that the two spaces correspond to semantics and syntax. However, this seems in no way guaranteed or enforced in the current model. For instance, on a different dataset, it could entirely be possible that the embedding spaces capture different aspects of polysemy.  However, this is a central point of the paper and would require a more thorough analysis, either by a theoretical motivation or a more comprehensive evaluation across multiple datasets.

- In its current form, I found the experimental evaluation not convincing. The qualitative analysis of filler embeddings is indeed interesting and promising. However, the comparisons to baseline models is currently lacking. For instance, perplexity results are far from state of the art and more importantly below serious baselines. For instance, the RNN+LDA baseline from Mikolov (2012) achieves already a perplexity of 92.0 on PTB (best model in the paper is 92.4). State-of-the-art models acheive perplexities around 50 on PTB. Without an evaluation against proper baselines I find it difficult to accurately assess the benefits of these models. While language modeling in terms of perplexity is not necessarily a focus of this paper, my concern translates also to the remaining experiments as they use the same weak baseline.

- Related to my point above, the experimental section would benefit significantly if the paper also included evaluations on downstream tasks and/or evaluated against existing methods to incorporate structure in language models.

Overall, I found that the paper pursues interesting and promising ideas, but is currently not fully satisfying in terms of evaluation and discussion.

---

> ### Author Response · Authors · 2018-11-08
> **Response to reviewer 2: Part 1**
>
> First, we would like to thank you for the kind words regarding our general idea. We fully acknowledge the validity of the many concerned raised here. We provide responses to them below:
>
> (1) “From a modeling perspective, the paper seems to conflate two points”:
>
> First of all, we agree that there are two points to be made here as you have pointed out:
> (a) the potential benefit of a role-filler approach
> (b) the architectural or computational advantage of any specific instance of such an approach.
>
> In writing this paper, we have the following considerations. First, we use language modeling as our testbed to investigate (a). As we explained in the intro, “the versatility of language modeling [as a complementary or pretraining task] demonstrates that some linguistic regularities much be present”. The recent success of BERT [1] and ELMO [2] across many tasks (including some very linguistics-oriented benchmarks) reflects this point as well. This being said, we acknowledge there are many other tasks that could be used to investigate (a) -- for instance, [3] used QA as the main task, and we personally thought about summarization on the ground that a summary has a clear designation of sentential roles (e.g., event name, location, etc). However, the simplicity of LM, coupled with its minimal necessity for supervision, convinced us to focus on LM instead.
>
> Second, while there are many other instances of variable-binding framework (TPR being one of the most prominent examples), we decided to investigate into HRR on computational grounds. This was explained in our background section (“makes HRR a more practical choice”). We will elaborate on this point more in the updated version.
>
> Our claim in the paper is based on the two considerations above. We believe that both (a) and (b) should be investigated fully, but given that this is our initial attempt, we think it’s reasonable to make some simplifying assumptions.
>
> We hope this address your concern.
>
> (2) “It is also not clear to me in what way we can interpret the different filler embedding”
>
> We agree that the separation of semantics and syntax is not guaranteed. However, nor did we claim it to be. We stated in the intro that our model can “effectively separate certain aspects of word or chunk representation, which roughly corresponds to a division between syntax and semantics”. The vagueness of our statement is precisely due to the fact that our model doesn’t have a “syntax training loss” or “semantics training loss”. In light of this, we argue that it is interesting and somewhat surprising that HRR-enabled models learned to separate these two aspects without a dedicated loss term. This goes to show that an inductive bias can be beneficial.
>
> We also agree that we need to make more comprehensive evaluation. We are currently expanding our experiments to more datasets (wiki, one-billion-word, possibly some domain-specific texts, or some subset of them), and we will provide an updated version as soon as possible. On the other hand, we would like to point out that for all our experiments on PTB, we observed a consistent pattern that the first role (the one without downweighting the dot product at the start of training) always corresponds more to syntax than semantics, regardless of hyperparameter setting or random seed. We think it’s because syntactic cues/signals (e.g., POS tags) are relatively easier to identify than semantics ones (e.g., topic relatedness), and therefore the first set of embeddings tend fo consistently capture the more syntactic aspect. Of course, this pattern will carry more weight if our new round of experiments also confirm it.
>
> (3) “In its current form, I found the experimental evaluation not convincing”
>
> We are fully aware that our baseline seems to underperform, as pointed out by reviewer 3 as well. First we would like to point out that contrary to common practice in LM literature, “we do not assume that the contiguous sentences in the raw data are fed sequentially as input”, and as a result “we do not initialize the hidden state of LSTM with the last state from the last batch” (section 4.1). We took this approach to ensure that chunk-level representations capture only intra-sentential roles -- we do not consider discourse-level features. The downside of this is that we can no longer reply on information from the last sentence to help predict the current one.
>
> Meanwhile, we are also running another word-level baseline that follows the common practice in LM literature. We will update the results shortly.

---

> ### Author Response · Authors · 2018-11-08
> **Response to reviewer 2: Part 2**
>
> (4) “the experimental section would benefit significantly if the paper also included evaluations on downstream tasks and/or evaluated against existing methods to incorporate structure in language models.”
>
> As for downstream task, due to space limit, it’s hard to fully investigate the potential benefits besides the decomposed representations which we spent most of our experimental section on. However, we are planning on running a POS tagging task using learned representations as features for a linear classifier. We are also planning on running the model on SRL task. We believe these tasks would be good testbeds for our proposed method, and also address your concern here.
>
> As for comparison against existing methods, we are not aware of any directly applicable approach. There are certainly many existing methods that try to incorporate structures, but mostly to enhance their representation, not decompose their representation. Moreover, the unsupervised nature of our approach makes direct comparison even harder. Of course, we can be totally ignorant, and we would appreciate any advice from you if you are aware of any specific comparable approach that fits the scenario here.
>
> [1] Devlin et al., BERT: Pre-training of Deep Bidirectional Transformers for Language Understanding
> [2] Peters et al., Deep contextualized word representations
> [3] Huang et al., Tensor Product Generation Networks for Deep NLP Modeling

---

### Author Response · Authors · 2018-11-08
**Thank you all for your reviews!**

We thank all the reviewers for their insightful comments and suggestions. We are aware of the general concern about the underperformance of baseline LMs. We will provide an updated submission shortly, with extended experimental analysis and improved baselines.

---

### Author Response · Authors · 2018-11-27
**Revision available now**

We made some major revisions to our initial submission. In particular, we significantly substantiated our experimental results and analysis. Here are a few key take-aways from our revision:
1. We retrained our models and improved the baseline performance on PTB. Perplexity is now on par with existing literature.
2. In addition, we trained models on the One-Billion-Word benchmark data with various portions of the training set and reported results.
3. We ran extensive experiments to both intrinsically and extrinsically evaluate the HRR models:
	- For intrinsic evaluation, we conducted experiments on 18 word embedding benchmark datasets and show improvements over the baseline in almost all cases.
	- For extrinsic evaluation, we conducted experiments on 6 downstream tasks and also show consistent improvements.
4. We conducted experiments to quantitatively evaluate chunk-level HRR embeddings, and showed that although trained with weak supervision, chunk embeddings do correspond to some gold semantic role labels.

We have also made other revisions addressing your questions and concerns. We hope you could take a look at our revised submission, and any further feedbacks and discussions are always welcome.

---

### Comment · AnonReviewer3 · 2018-12-06
**Revision provides new results and analyses**

I have read the new revision, and noted the following positive points:
1. It adds results on the 1B word dataset.
2. It reports improved perplexity numbers for both the baseline and the proposed models.
3. It evaluates the quality of the word embeddings via many benchmark datasets.
4. It adds an evaluation of chunk embeddings via cluster analysis w.r.t semantic roles.
* All these are useful quantitative analyses that I thought were missing from the previous version.

One concern that I have is that most of the results are not compared to the SOTA. I don't think it's necessary to beat the SOTA, but at least providing numbers would put results in perspective. At this point I am not sure if the baseline is indeed good enough (e.g., on PTB language modeling there are vanilla LSTM models that get <60, while the paper reports a baseline of 76). The same is true for the results in the word level analysis (section 3.3).

There are also a few lingering questions for the authors from my comment from Nov 17. Please take a look at that.

---

> ### Author Response · Authors · 2018-12-14
> **Regarding baseline results**
>
> We thank you for reviewing our updated submission. For reproducibility, our implementation of the baseline LM is based off the open-source Tensorflow implementation of RNNLM tutorial: https://github.com/tensorflow/models/blob/master/tutorials/rnn/ptb/ptb_word_lm.py  (we plan to  open-source our codes for the paper as well, and they will be easily reproducible due to compatibility with the public Tensorflow implementation). The perplexity results on PTB are reported in the official document of the Tensorflow implementation, which we quote as follows:
>
> ===========================================
> | config | epochs | train | valid  | test
> ===========================================
> | small  | 13     | 37.99 | 121.39 | 115.91
> | medium | 39     | 48.45 |  86.16 |  82.07
> | large  | 55     | 37.87 |  82.62 |  78.29
>
> Our baseline perplexity results on PTB reported in the revised submission is just about the same as the "large" setting in the open-source implementation. We are aware that these numbers are not close to the SOTA results, but we did not adopt any of the sophisticated model modifications or training techniques from those SOTA models (for example, [1] reports PTB perplexity 56.8 using fraternal dropout, [2] reports PTB perplexity 57.3 using weight-dropped LSTM + averaged SGD). Our intention was to build the HRR-LM based upon a simple, plain RNNLM, from which the effect of enabling the HRR mechanism can be best studied without introducing too much modeling or training complication. We do acknowledge that there is still much room left for perplexity improvement, for both the baseline and our proposed HRR LMs.
>
> Please also find our response to your earlier comments below.
>
> [1] Fracternal Dropout, Konrad Zołna et. al, ICLR 2018
> [2] Regularizing and Optimizing LSTM Language Models, Stephen Merity et. al, arXiv:1708.02182v1

---

### Meta-Review · Area_Chair1 · 2018-12-14
**Interesting task, but a lack of satisfaction with the results**

**Confidence:** 2
**Recommendation:** Reject

**Metareview:**

This paper proposes the use of holographic reduced representations in language modeling, which allows for a cleaner decomposition of various linguistic traits in the representation. Results show improvements over baseline language models, and analysis shows that the representations are indeed decomposing as expected.

The main reviewer concern was the lack of strength of the baseline, although the authors stress that they were using the default baseline from TensorFlow, which seems like it will be reasonable to me. Another concern is that there is other work on using HRR to disentangle syntax and semantics in representations for language (e.g. "Distributed Tree Kernels" ICML 2012, but also others), that has not been considered.

Based on this, this seems like a very borderline case. Given that no reviewer is pushing strongly for the paper I'm leaning towards not recommending acceptance, but I could very easily see the paper being accepted as well.

---

> ### Author Response · Authors · 2018-12-21
> **Thank you all for your comments. Some more words.**
>
> We thank all reviewers and chair for your comments. While we fully understand your concerns regarding the baseline results and related work, we hope to make it clear (once more) that:
> 1) Our intention was to make the results easily reproducible based on the widely available Tensorflow open-source implementation of LM on public datasets, with no sophisticated tricks or model modifications involved. At the same time, we strongly believe that the main contribution of our submission - decomposition of representation - does not have to be correlated with perplexity results. Better perplexity results do not guarantee decomposed representation, nor is a good decomposed representation hinged on good perplexity results.
> 2) We acknowledge Reviewer 1 for pointing out related works, however the existing approaches, although using HRR as a component, are very different from the ones we proposed in our paper. They would not naturally apply to learning disentangled linguistic features, the problem we aim to tackle and major contribution we make in our paper. Our understanding is that Reviewer 1 raised this point in debating with our claim that a naively implemented chunk-level model is intractable, and it was not his intention to directly apply the work on tree kernel HRR to the decomposition task at hand.
>
> We believe that our proposed model, formulated specifically for addressing the topic of disentangled linguistic representation, is novel and effective, and provides a viable approach for future research. We would greatly appreciate it if our comments and previous responses are taken into more serious consideration, and decisions properly revised.